# The eukaryotic replisome tolerates leading-strand base damage by replicase switching

Thomas A Guilliam [iD] & Joseph TP Yeeles[*] [iD]

## Abstract

The high-fidelity replicative DNA polymerases, Pol ε and Pol δ, are generally thought to be poorly equipped to replicate damaged DNA. Direct and complete replication of a damaged template therefore typically requires the activity of low-fidelity translesion synthesis (TLS) polymerases. Here we show that a yeast replisome, reconstituted with purified proteins, is inherently tolerant of the common oxidative lesion thymine glycol (Tg). Surprisingly, leading-strand Tg was bypassed efficiently in the presence and absence of the TLS machinery. Our data reveal that following helicase–polymerase uncoupling a switch from Pol ε, the canonical leading-strand replicase, to the lagging-strand replicase Pol δ, facilitates rapid, efficient and error-free lesion bypass at physiological nucleotide levels. This replicase switch mechanism also promotes bypass of the unrelated oxidative lesion, 8-oxoguanine. We propose that replicase switching may promote continued leading-strand synthesis whenever the replisome encounters leading-strand damage that is bypassed more efficiently by Pol δ than by Pol ε.

**Keywords** DNA damage tolerance; DNA polymerase; DNA replication; replisome; translesion synthesis

**Subject Category** DNA Replication, Recombination & Repair

**The EMBO Journal (2021) 40: e107037**

## Introduction

Unrepaired DNA lesions encountered during genome duplication can stall the eukaryotic replicases, polymerase (Pol) α, δ and ε (Zeman & Cimprich, 2014). Since lagging-strand synthesis is discontinuous, stalling of Pol δ, the principal lagging-stand replicase (Burgers & Kunkel, 2017), is overcome through priming of the next Okazaki fragment by Pol α, leaving a single-stranded (ss) DNA gap (Taylor & Yeeles, 2018). In contrast, leading-strand synthesis is mainly continuous and stalling of Pol ε, which associates with the replicative helicase CMG (Cdc45-MCM-GINS) for bulk replication (Pursell *et al*, 2007; Langston *et al*, 2014; Sun *et al*, 2015), causes uncoupling. Here, template unwinding and lagging-strand replication continue at a decreased rate in the absence of leading-strand synthesis (Taylor & Yeeles, 2018, 2019). In higher eukaryotes, repriming by a second primase, PrimPol, may restart leading-strand synthesis downstream of damage (Guilliam & Doherty, 2017). However in *Saccharomyces cerevisiae*, which lacks PrimPol, leading-strand repriming by Pol α is inefficient (Taylor & Yeeles, 2018).

Another mechanism to restart stalled leading strands, that can also fill in ssDNA gaps, involves recruitment of translesion synthesis (TLS) DNA polymerases that can directly bypass damage (Marians, 2018). TLS polymerases are error-prone and must be strictly regulated to prevent mutagenesis (McCulloch & Kunkel, 2008). *Saccharomyces cerevisiae* possess three TLS polymerases, Pol η, Pol ζ and the deoxycytidyl transferase, Rev1 (Waters *et al*, 2009), of which Pol ζ is responsible for the majority of spontaneous and induced mutagenesis (Gan *et al*, 2008). Using an origin-dependent eukaryotic DNA replication system, reconstituted with purified budding yeast proteins (Yeeles *et al*, 2015, 2017), we recently showed that Pol δ binds the stalled leading strand upon uncoupling at a cyclobutane pyrimidine dimer (CPD). Here, monoubiquitination of PCNA by Rad6–Rad18 stimulated a switch to Pol η to promote lesion bypass (Guilliam & Yeeles, 2020a). Rev1 and Pol ζ were absent from these experiments, and therefore, the full interplay between TLS polymerases and the replisome could not be investigated.

In this study, we initially sought to investigate how the complete *S. cerevisiae* TLS machinery interfaces with the replisome to facilitate leading-strand lesion bypass by focusing on thymine glycol (Tg), the most common oxidative product of thymine, which forms ~ 300 times per cell each day in humans (Breimer & Lindahl, 1985; Adelman *et al*, 1988). Tg has been reported to stall both prokaryotic and eukaryotic replicases, but be efficiently bypassed by yeast Pol ζ *in vitro* (Clark & Beardsley, 1987; Johnson *et al*, 2003). Because Rev1 is a key binding partner of Pol ζ that recruits it to monoubiquinated PCNA (Martin & Wood, 2019), we reasoned that monitoring Tg bypass would allow us to delineate the interplay of both additional factors with the replisome. Unexpectedly, however, our work has instead revealed that the yeast replisome is inherently tolerant of leading-strand Tg. Rapid, efficient and error-free lesion bypass occurred independently of the TLS machinery via a replicase switch mechanism.

## Results

### The replisome is inherently tolerant of a leading-strand Tg

To analyse the contribution of the full *S. cerevisiae* TLS machinery to tolerance of a leading-strand Tg, we used the reconstituted

Division of Protein and Nucleic Acid Chemistry, Medical Research Council Laboratory of Molecular Biology, Cambridge, UK
*Corresponding author. Tel: +44 01223 267163; E-mail: jyeeles@mrc-lmb.cam.ac.uk

system previously described (Yeeles *et al*, 2015, 2017; Taylor & Yeeles, 2018; Guilliam & Yeeles, 2020a) and a linear 9.7 kb template with a single Tg located ~ 3 kb from the origin in the leading-strand template (Fig 1A). Here, MCM double hexamers loaded at the origin are activated to form two replisomes, the leftward fork encounters the Tg, whilst the rightward fork generates an ~ 1.5 kb leading-strand product (run off) (Fig 1A). If lesion bypass occurs, full-length products containing 8.2 kb leading strands are generated. However, if leading-strand synthesis stalls for a prolonged period at the Tg, uncoupled forks and uncoupled products will be generated (Fig 1B). Assays were performed in the presence of the TLS machinery—Pol η, Pol ζ, Rev1 (Fig EV1) and factors required for PCNA ubiquitination, Rad6–Rad18, Uba1 and ubiquitin—on undamaged (UD) and Tg templates, as well as a CPD-containing template that requires TLS by Pol η for complete leading-strand replication (Guilliam & Yeeles, 2020a).

On all three templates, full-length products comprising 8.2 kb leading strands (FL-lead) accumulated during the experiment, demonstrating significant bypass of both the Tg and CPD lesions. Compared to the UD template, synthesis of FL-lead was delayed on the CPD template and to a lesser, but still notable, extent on the Tg template (Fig 1C, native lanes 1, 4 and 7) and stalled leading strands of ~ 3 kb (stall) were observable at 15 min (Fig 1C, denaturing lanes 4 and 7). Therefore, although both lesions were efficiently bypassed, replisome progression was delayed, with replication past Tg occurring more rapidly than past the CPD under these conditions.

To assess the requirement of the TLS machinery for Tg bypass, the experiment was repeated in the absence of these factors (Fig 1D). As expected, full-length products containing FL-lead were again rapidly synthesised on the UD template (Fig 1D, lanes 1–3), whereas no FL-lead was synthesised on the CPD template. Here, stall persisted throughout the reaction, resulting in the accumulation of uncoupled forks and the appearance of an uncoupled product after 30 min (Fig 1B and D, lanes 7–9).

Remarkably, despite the omission of the TLS machinery, full-length products containing FL-lead were efficiently synthesised from the Tg template. In fact, Tg bypass and the completion of leading-strand replication occurred with similar efficiency in the presence and absence of the TLS machinery (compare Fig 1C lanes 4–6 and 1D lanes 4–6). This demonstrates that the TLS machinery is not required for bypass of a leading strand Tg by the replisome. Moreover, it reveals an intrinsic tolerance of this lesion by at least one of the replicases.

## Tg causes uncoupling of leading-strand synthesis

The rapid and efficient synthesis of fully replicated leading strands from the Tg template, and the absence of detectable uncoupled products, indicated that CMG-bound Pol ε (CMGE) might catalyse processive Tg bypass without the uncoupling of leading-strand synthesis from template unwinding. Here, the slight delay in full-length product generation and small amount of stall in Fig 1D may be due to failure of a subset of CMGE complexes to efficiently bypass Tg or slowing of CMGE progression during lesion bypass. Previously, to detect transient uncoupling during bypass of a CPD we used oligonucleotides that anneal to the leading-strand template downstream of damage and promote leading-strand restart (Guilliam & Yeeles, 2020a). Here, if uncoupling occurs, the

oligonucleotide binds to exposed ssDNA and is extended to generate a discontinuous leading-strand restart product at the expense of FL-lead. Conversely, if lesion bypass is tightly coupled to template unwinding, production of FL-lead should be refractory to competition from the oligonucleotide.

To examine whether transient uncoupling occurs during Tg bypass, we used oligonucleotides mapping 21 nt and 265 nt downstream of the lesion and compared them to a scrambled oligonucleotide (S) that does not promote restart (Fig 2A). On the UD template, FL-lead was readily synthesised in the presence of all three oligos (Fig 2B, lanes 1–3), confirming that oligonucleotide-mediated restart does not occur without uncoupling. As we observed previously, the 21 nt oligonucleotide reduced the overall efficiency of replication, but did not qualitatively affect the reaction products (Guilliam & Yeeles, 2020a).

On the Tg template, FL-lead was efficiently synthesised in the presence of the scrambled oligonucleotide, again demonstrating Tg bypass independent of the TLS machinery (Fig 2B, lane 4). In contrast, addition of both the 21 nt and 265 nt oligonucleotides reduced the accumulation of FL-lead and discontinuous restart products of the expected sizes (Fig 2A) were observed (Fig 2B, lanes 5 and 6, and Fig 2C). However, whereas oligo restart products predominated with the 21 nt oligonucleotide, FL-lead was still the major reaction product with the 265 nt oligonucleotide (Fig 2C). This indicates that transient uncoupling of leading-strand synthesis from DNA unwinding occurred during the majority of Tg bypass events, but was usually restricted to less than 265 nt beyond the lesion.

That some FL-lead was still clearly visible with the 21 nt oligonucleotide indicated that either a subset of Tg lesions were bypassed processively by CMGE or uncoupling was sometimes too transient to permit oligonucleotide binding and extension before bypass. Pol δ binds the nascent leading strand upon uncoupling (Guilliam & Yeeles, 2020a). We therefore reasoned that a Pol δ mutant lacking polymerase activity (Pol δ^cat, Pol3 D608A) (Aria & Yeeles, 2019) could be used to trap uncoupled leading strands to determine whether uncoupling is a universal response to leading-strand Tg. To monitor fork progression more synchronously, a pulse chase experiment was performed on UD and Tg templates in the absence of Pol δ. Pol δ^cat was added at the start of the chase to prevent inhibition of initiation (Aria & Yeeles, 2019; Fig 2D). On the UD template, FL-lead was generated both in the absence and presence of Pol δ^cat (Fig 2D, lanes 1–6), confirming the mutant does not inhibit coupled leading-strand synthesis. On the Tg template, no FL-lead products were observed in the presence of Pol δ^cat and stall persisted across the time course, revealing that uncoupling occurred at most, if not all, replication forks (Fig 2D, lanes 10–12).

## Leading-strand Tg bypass requires RFC/PCNA

Because leading-strand Tg is not bypassed processively by CMGE, a non-CMG associated or "free" replicase must be responsible for Tg bypass. Efficient bypass by free Pol ε or Pol δ is likely to require PCNA and the clamp loader RFC since PCNA is a processivity factor for both polymerases (Chilkova *et al*, 2007). We therefore compared Tg bypass in the absence and presence of RFC/PCNA. In the presence of RFC/PCNA, FL-lead products were evident by 10 min and continued to accumulate across the time course (Fig 2E, lanes

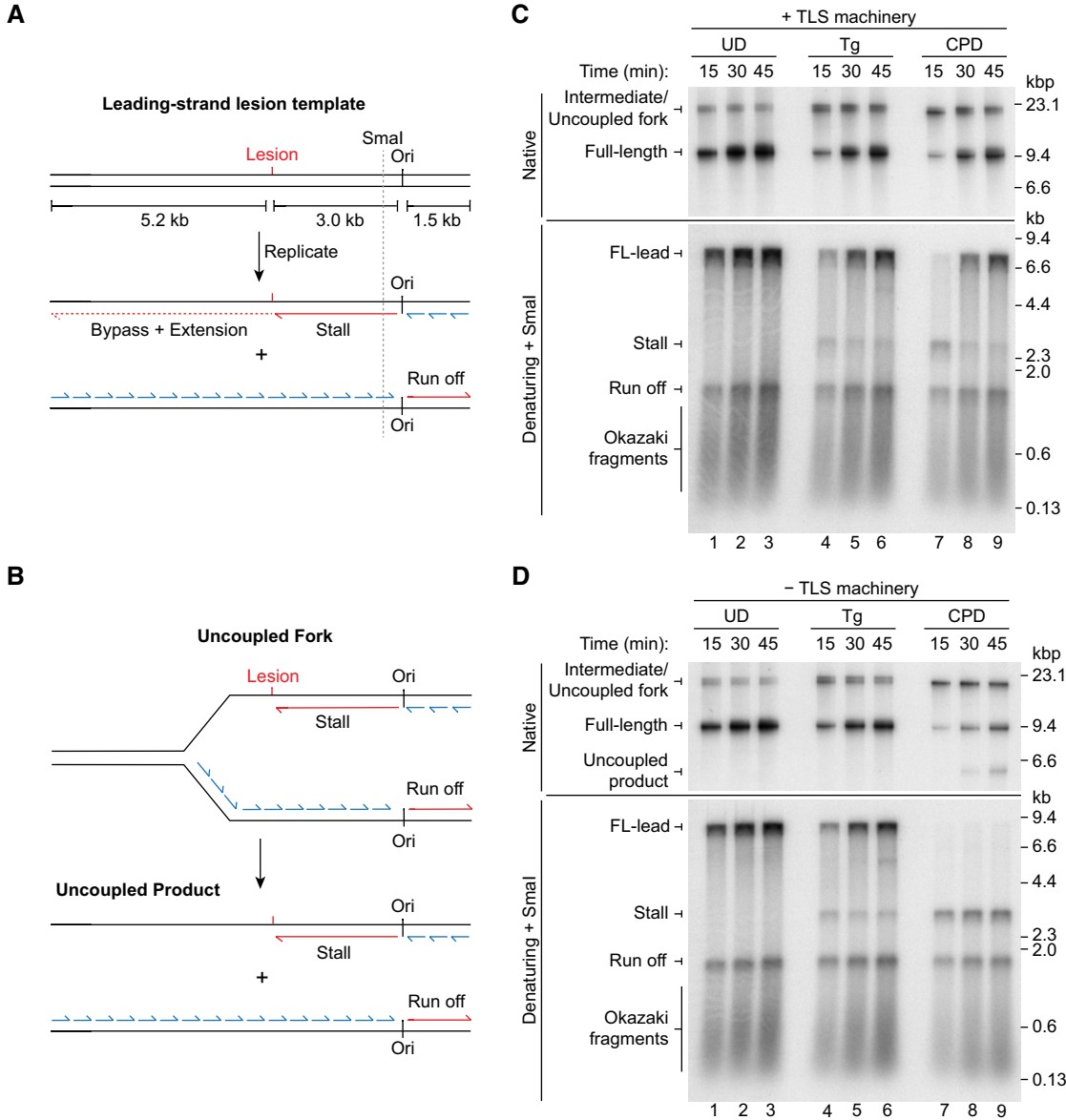

**Figure 1. The replisome is inherently tolerant of a leading-strand Tg lesion.**

A   Schematic of the ARS306 leading-strand lesion template showing the origin of replication (Ori) and leading- (red) and lagging-strand (blue) replication products. The SmaI restriction site is shown; cleavage is used to reduce heterogeneity in product sizes for denaturing gel analysis.

B   Schematic of the uncoupled fork and uncoupled product generated by persistent stalling of leading-strand synthesis.

C, D   Standard replication assays on undamaged (UD), thymine glycol (Tg) and cyclobutane pyrimidine dimer (CPD) templates in the presence (C) and absence (D) of Pol η, Pol ζ, Rev1, Rad6–Rad18, Uba1 and ubiquitin (TLS machinery). Reactions contained 2.5 nM Pol δ.

7–12). Although leading strands were extended to the stall position in the absence of RFC/PCNA, no FL-lead was generated, even after 60 min (Fig 2E, lanes 7–12), demonstrating that bypass of Tg had not occurred. This requirement of RFC/PCNA for complete replication of the Tg template suggests that free Pol ε or Pol δ facilitates uncoupled tolerance of the lesion.

## Pol δ facilitates rapid and efficient leading-strand Tg bypass

Although Pol ε is the canonical leading-strand polymerase for bulk DNA replication (Burgers & Kunkel, 2017), Pol δ also participates in leading-strand synthesis during initiation and recoupling after CPD bypass by Pol η (Guilliam & Yeeles, 2020b). It is therefore possible that one or both of these replicative polymerases facilitates Tg bypass by the replisome. Indeed, Tg bypass was observed in the absence of Pol δ (Fig 2D, lanes 7–9), but this appeared to be less efficient than in reactions performed in its presence (Fig 1C, lanes 4–6 and 2D, lanes 7–12). Furthermore, Pol δ$^{cat}$ inhibited Tg bypass (Fig 2D, lanes 10–12), which might indicate that Pol δ binds preferentially to the nascent leading strand after it is released by CMGE, as we have demonstrated occurs at a CPD (Guilliam & Yeeles, 2020a).

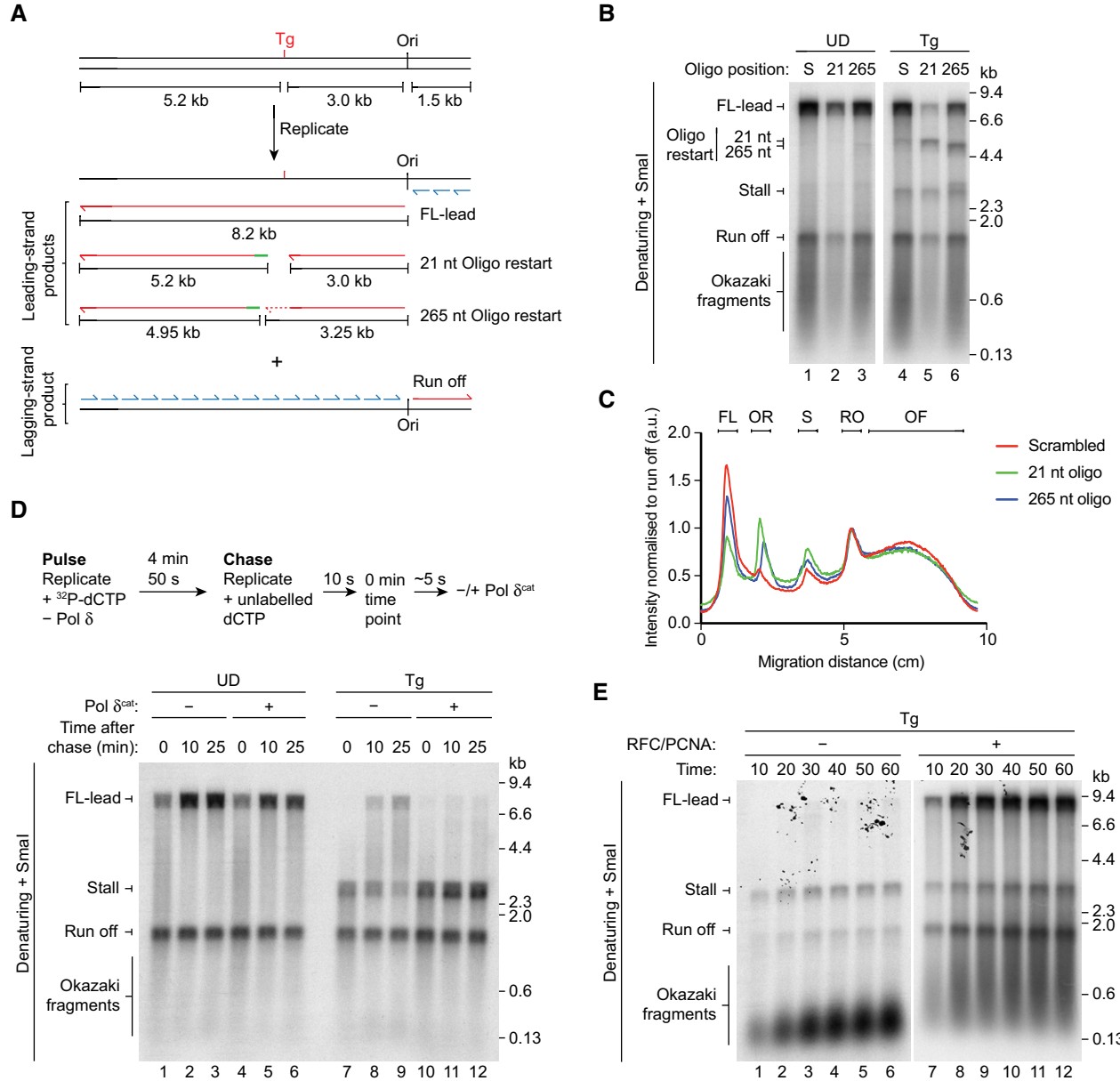

**Figure 2. A leading-strand Tg causes uncoupling.**

A  Schematic showing the possible replication products and oligonucleotide restart products for the assay in (B). Green lines represent the restart oligonucleotides and the dashed line indicates extension of the stall product which will occur due to bypass of the Tg after oligonucleotide binding.

B  Oligonucleotide competition assay on UD and Tg templates. S; scrambled oligonucleotide.

C  Lane profiles of lanes 4–6 in (B) normalised to the run off product. FL; FL-lead, OR; oligonucleotide restart, S; Stall, RO; Run off, OF; Okazaki fragments.

D  Pulse chase reaction on UD and Tg templates performed as shown. Where present, Pol δ$^{cat}$ (10 nM) was added directly after addition of the chase.

E  Standard replication reaction performed in the absence and presence of RFC/PCNA on the Tg template.

To evaluate the contributions of Pol ε and Pol δ to Tg bypass, we performed a pulse chase experiment in which Pol δ was absent from the pulse and either added or omitted during the chase (Fig 3A). In the absence of Pol δ, only limited FL-lead was generated 15 min after addition of the chase, with stall the predominant replication product at each time point (Fig 3B, lanes 1–5). In comparison, when Pol δ was present in the chase, stall was efficiently extended to FL-lead, with almost complete bypass by 15 min (Fig 3B, lanes 6–10).

A similar result was observed when $^{32}$P-dTTP and elevated dTTP were used in the pulse and chase, respectively, showing that bypass was not specifically caused by the increased dCTP concentration in the chase (Fig EV2A). Figure EV2B demonstrates that the increase in FL-lead that was observed upon addition of Pol δ was not due to TLS performed by low levels of contaminating Pol ζ, because Pol δ purified from a *REV3* (catalytic subunit of Pol ζ) deletion strain also substantially stimulated FL-lead production.

The data in Fig 3B suggest Pol δ is the replicase primarily responsible for Tg bypass. However, given Pol δ is crucial to recouple leading strands following CPD bypass by Pol η (Guilliam & Yeeles, 2020a), the stall product observed in the absence of Pol δ might have advanced a short distance beyond Tg without being properly extended. Such activity would not be resolved due to the limited resolution of the denaturing agarose gel in Fig 3B. Indeed, at 15 min there appeared to be a smear above the stall position in the reaction lacking Pol δ, indicative of inefficient recoupling (Fig 3B, lane 5) (Guilliam & Yeeles, 2020a).

To confirm Pol δ was directly facilitating Tg bypass, we performed another pulse chase experiment and digested the reaction products with SwaI and BamHI for analysis on urea polyacrylamide gels. SwaI and BamHI map upstream and downstream of the Tg, respectively, such that cleavage of stall generates a 165 nt product with the 3′-end defined by stalling at the lesion, whereas bypass and extension past the BamHI site generates a 187 nt bypass product following cleavage (Fig 3C). BamHI cleavage is staggered allowing resolution of bypass products from digested lagging strands (Cut lag). Consistent with the data in Fig 3B, in the absence of Pol δ, bypass was generated; however, significant stall still remained by the final 9-min time point (Fig 3D, lanes 1–7). This indicates that Pol ε can indeed perform Tg bypass; however, it does so slowly and inefficiently. When Pol δ was added, almost complete extension of stall past the BamHI site occurred within 1.5 min and bypass continued to accumulate through the reaction (Fig 3D, lanes 8–14). These experiments reveal that Pol δ, the canonical lagging-strand replicase, not only accesses the leading strand following uncoupling, but facilitates rapid and efficient Tg bypass.

To further examine the efficiency of Tg bypass in the presence of Pol δ, we performed a titration of the enzyme in a pulse chase experiment where it was added immediately after the chase (after the 0-min time point) (Fig 3E). Pol δ stimulated conversion of stall to bypass across the titration range. Quantification revealed that 0.63 nM Pol δ was sufficient to substantially enhance Tg bypass, with only a slight further increase at higher concentrations (Fig 3F). This demonstrates that sub-nanomolar concentrations of Pol δ promote rapid and efficient leading-strand Tg bypass.

In *S. cerevisiae*, DNA damage increases dNTP levels, which has been proposed to enhance the efficiency of lesion bypass by DNA polymerases (Chabes *et al*, 2003). Since Pol δ is stimulatory, but not essential, for leading-strand Tg bypass (Fig 2D), we considered that elevated dNTP levels might promote Tg bypass by Pol ε, perhaps limiting the involvement of Pol δ. Comparison of lesion bypass at 30 and 150 μM dNTPs in a pulse chase experiment revealed that increasing dNTP concentrations stimulated the production of FL-lead in the absence of Pol δ (Fig EV2C, compare lanes 1–3 and 7–9). However, Pol δ still enhanced FL-lead production at the elevated dNTP concentration (compare lanes 7–9 and 10–12). Furthermore, Fig EV2D shows that addition of Pol δ$^{cat}$ during the chase phase of a pulse chase reaction almost completely inhibited the production of FL-lead at both dNTP concentrations, demonstrating that elevated dNTPs do not prevent transient uncoupling during lesion bypass at most, if not all, replication forks. These results, together with the observation that Pol δ outcompetes Pol ε for uncoupled 3′-ends (Guilliam & Yeeles, 2020a), indicate that Pol δ plays a major role in leading-strand Tg bypass even when dNTP levels are elevated.

## Pol δ and Pol ε bypass Tg in primer extension assays

The efficiency of Tg bypass in our replication reactions was somewhat surprising given that yeast Pol δ was previously reported only to be able to incorporate nucleotides opposite, and not extend from, Tg in a primer extension assay lacking accessory factors (Johnson *et al*, 2003). We therefore considered that during Tg bypass in the replication assay, Pol δ might promote nucleotide incorporation opposite the lesion, with Pol ε then performing the extension step. We therefore investigated Tg bypass by Pol δ and Pol ε during primer extension assays in the absence of other polymerases and replication proteins. Since PCNA is a processivity factor for both polymerases (Chilkova *et al*, 2007), both it and the clamp loader RFC were included. DNA synthesis was monitored on a 50 nt template with a 20 nt primer annealed to place the Tg 11 nt downstream of the 3′-end of the primer ($N + 11$) (Fig 3G).

At the lowest Pol δ concentration (1 nM), stalling was observed at the base immediately preceding the Tg ($N + 10$); however, some incorporation opposite the lesion ($N + 11$) and full extension ($N + 27$–30) were apparent (Fig 3G, lane 2). This demonstrates that even when present at a 10-fold lower concentration than the template, Pol δ can incorporate opposite and extend from the Tg to some degree. As the concentration of the enzyme increased, the $N + 11$ stall readily decreased and full extension became the predominant reaction product, with almost no stalling observed at the highest concentration (Fig 3G, lanes 2–6). Additionally, Tg bypass by Pol δ was consistent across a range of potassium glutamate concentrations (Fig EV3A) and, although RFC/PCNA were stimulatory, they were not required to observe bypass (Fig EV3B). Collectively, these data reveal that Pol δ alone can bypass a Tg lesion.

In comparison, stalling at $N + 10$ was more apparent across the Pol ε titration (Fig 3G, lanes 7–12). Some $N + 11$ and $N + 12$ incorporation occurred; however, only a faint $N + 30$ product was generated even at the highest Pol ε concentrations. Although RFC/PCNA stimulated the overall activity of Pol ε in these experiments, they did not prevent stalling at Tg (Fig EV3C). Similarly, bypass by Pol ε was not significantly altered when a 30 nt primer was used (Fig EV3D). To understand whether the exonuclease activity of Pol ε was limiting Tg bypass, we performed a primer extension assay with an exonuclease-deficient variant of Pol ε (Pol ε$^{exo−}$) (Fig EV3E). Here, the predominant stall product in the Pol ε$^{exo−}$ reactions was shifted from the $N + 10$ observed with Pol ε (stalling immediately before the lesion) to $N + 12$ (stalling upon incorporation of a single nucleotide after the lesion). Although more full extension was observed with Pol ε$^{exo−}$, suggesting that Pol ε exonuclease activity limits Tg bypass to some extent, a substantial amount of stalling still occurred compared to reactions containing equivalent concentrations of Pol δ (compare Fig EV3B and E). Overall, these results show that Tg does not present a complete block to either replicase, but Pol δ is inherently more efficient than Pol ε at lesion bypass.

## Pol δ likely performs insertion and extension during Tg bypass

Although Fig 3G revealed Pol δ bypasses Tg with greater efficiency than Pol ε, Pol ε will initially encounter the leading-strand Tg, and therefore, it is possible that it incorporates opposite the lesion before uncoupling and subsequent extension by Pol δ (Fig 4A). Moreover, the ability of Pol ε to incorporate opposite the lesion may be

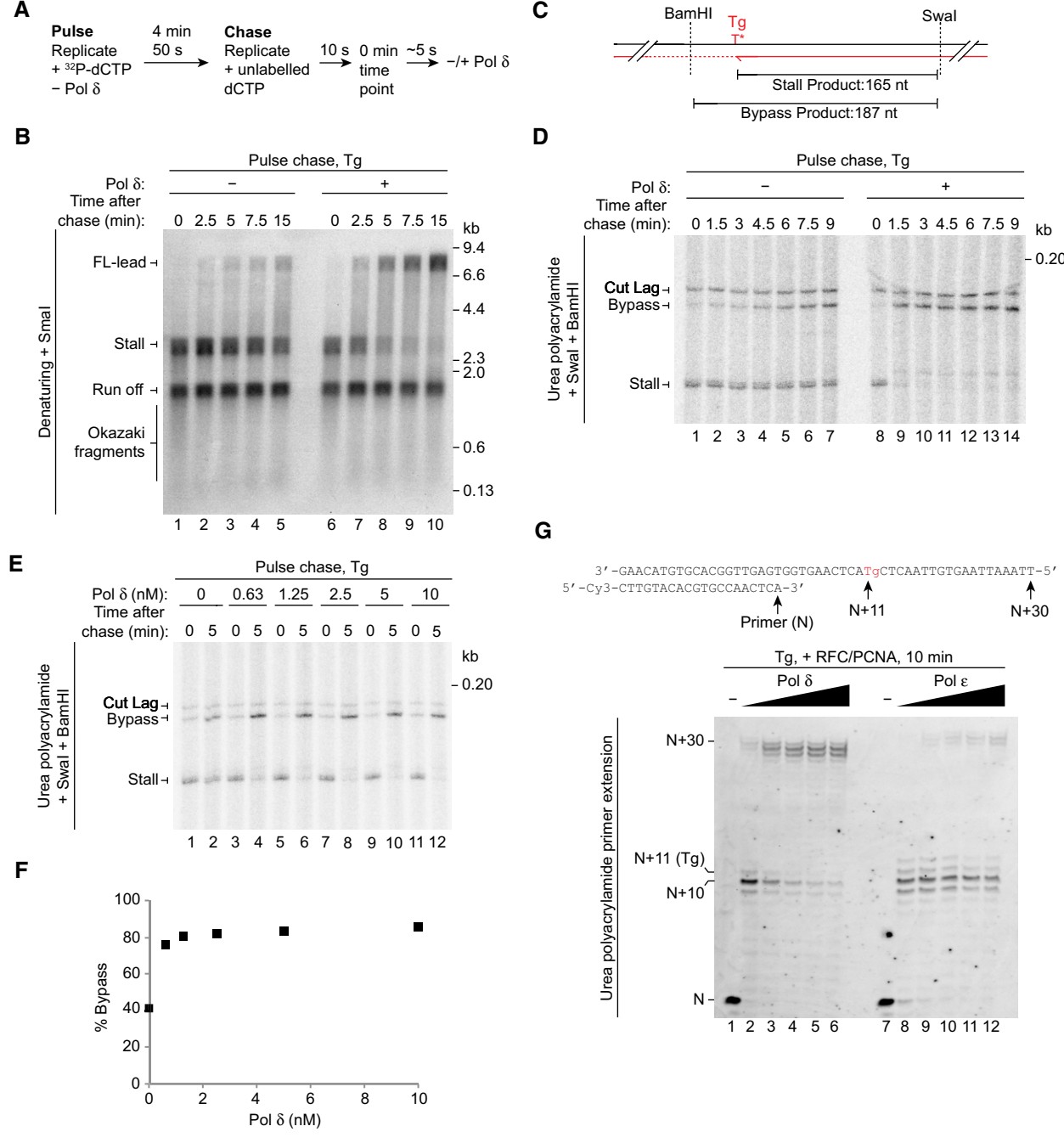

**Figure 3. Pol δ facilitates bypass of a leading-strand Tg.**

A  Outline of pulse chase reaction used in (B–E).

B  Pulse chase reactions on the Tg template in the absence and presence of 10 nM Pol δ analysed on a denaturing gel.

C  Schematic of stall and bypass products generated by SwaI and BamHI cleavage (D and E).

D  Pulse chase reaction on the Tg template in the absence and presence of 10 nM Pol δ analysed on a urea polyacrylamide gel.

E  Titration of Pol δ in a pulse chase reaction on the Tg template. Pol δ was added after the 0 min time point was taken.

F  Quantification of bypass 5 min after addition of the chase for the data in (E).

G  Titration of Pol δ or Pol ε (1, 2.5, 5, 7.5, 10 nM) into a primer extension assay on a Tg template in the presence of PCNA and RFC. The primer/template sequence and location of the lesion are shown above.

modulated when bound to CMG. To gain insight into which replicase incorporates opposite Tg, we mapped to single nucleotide resolution the position of the stalled leading strand. Reaction products

from pulse chase experiments where Pol δ was absent from the pulse were resolved following SwaI and BamHI cleavage and compared to a CPD reaction where stalling occurs at the base

immediately preceding the lesion (Guilliam & Yeeles, 2020a). Unlike the single stall product generated on the CPD template (Fig 4B, lanes 3 and 4), stalling at the Tg produced a doublet (Fig 4B, lanes 1 and 2). The more prominent lower band resolved at the same position as the CPD stall, indicating stalling immediately before the lesion. However, the second, less intense band resolved at the +1 position, corresponding to stalling after incorporation opposite Tg. We cannot determine if incorporation opposite Tg is performed by CMGE, or free Pol ε after uncoupling. In the latter case, it is likely that, when present, Pol δ would preferentially insert the base opposite the lesion, since it is immediately recruited to the uncoupled leading strand upon stalling (Guilliam & Yeeles, 2020a) and is more efficient at Tg bypass (Fig 3G). Indeed, the presence of Pol δ in the chase stimulated extension of both Tg stall products to bypass within 2.5 min (Fig 4B, compare lanes 5 and 6), but did not alleviate stalling at the CPD (Fig 4B, compare lanes 7 and 8). This reveals that Pol δ can perform both the insertion and extension steps of Tg bypass in the context of the replisome. Moreover, as the primary stall product generated in the absence of Pol δ is positioned immediately before the lesion, we consider it likely Pol δ is primarily responsible for insertion. However, we cannot rule out that insertion is performed by CMGE, as the lower stall product could be generated by removal of the base opposite Tg by the proofreading activity of free Pol ε after uncoupling (Fig EV3C).

### Tg bypass by the replisome is error-free

Unlike Pol ε and Pol δ, TLS polymerases are error-prone (McCulloch & Kunkel, 2008). Replicase-mediated Tg bypass would therefore have the advantage of preventing potentially error-prone DNA

synthesis surrounding the lesion site. However, for this to be an effective mechanism, nucleotide incorporation opposite the lesion must be faithful. Because our data does not exclude the possibility that Pol ε is responsible for at least a subset of insertions opposite Tg, we investigated the fidelity of both polymerases during lesion bypass. First, the nucleotide incorporated by each polymerase opposite Tg was tested in a primer extension assay. A 30 nt primer was annealed to the 50 nt Tg template, placing the lesion at the $N + 1$ position. Separate reactions were performed containing each individual nucleotide (dATP, dCTP, dGTP, dTTP). For both Pol δ and Pol ε, an $N + 1$ product was only observed with dATP (Fig 5A, lanes 2 and 7), indicating both polymerases insert the correct nucleotide opposite Tg.

Next, the fidelity of Tg bypass in a replication assay was assessed. To do this, we devised a method to isolate replicated leading strands. We reasoned that a nick in the leading-strand template would produce a truncated product that could be separated from duplex lagging-strand products and un-replicated parental DNA template. A BspQI site was therefore introduced ~ 4.9 kb downstream of the origin to enable nicking of the leading-strand template with Nt. BspQI prior to replication (Fig EV4A). Replication of a nicked UD template, both in the absence and presence of Pol δ, produced a lower molecular weight duplex leading-strand product of the expected size that was sufficiently resolved from the ~ 8.1 kb lagging-strand product (Fig EV4B, native). Denaturing gel analysis revealed that leading-strand synthesis stopped at the nick, generating a 4.8 kb truncated lead product as expected (Fig EV4B, denaturing).

To isolate bypass products for DNA sequencing, replication was performed on a nicked Tg template and leading strands were excised

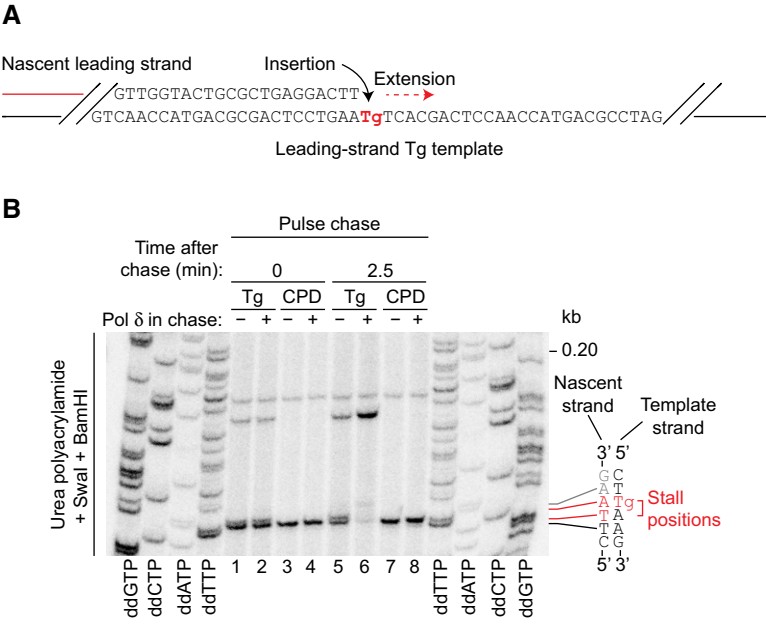

**Figure 4. Pol δ likely performs insertion and extension during Tg bypass.**

A  Schematic showing insertion and extension at the Tg site in the sequence context used in replication assays.

B  Mapping the 3′-end of the stalled nascent leading strand. Pulse chase reaction on Tg and CPD templates. Where present, Pol δ (10 nM) was added immediately after the 0 min time point. Products were cleaved with SwaI and BamHI and resolved alongside a sequencing ladder corresponding to the nascent strand. dd; dideoxy.

from a native agarose gel. Excised products were treated with Nt. BbvCI, which nicks the template strand on either side of the lesion, and DpnI, to digest any remaining template, before PCR amplification, sub-cloning and Sanger sequencing (Fig 5B). Three individual experiments were performed, two in the presence of Pol δ and one in its absence, producing 99, 97 and 95, sequencing reads, respectively. In all three experiments, bypass was error-free, except for a single deletion mutant opposite the lesion in each experiment (Fig 5C). Given the deletion occurred at the same rate in all experiments, both in the absence and presence of Pol δ, we expect it is due to the absence of Tg in a small percentage of the oligonucleotides used to make the template. Importantly, these results confirm that bypass of leading-strand Tg by Pol ε and Pol δ is predominantly error-free.

### Replicase switching promotes bypass of 8-oxoguanine

To test whether replicase switching promotes replisome-mediated bypass of other single-base lesions, we performed experiments on a template containing the common oxidative lesion 8-oxoguanine (8oxoG) in the leading strand. 8oxoG is structurally distinct from Tg

and should therefore provide insight into the generality of the replicase switch mechanism. Replication of the 8oxoG template was compared to UD and Tg templates in the absence of canonical TLS factors (Fig 6A). On all three templates, full-length duplex products containing FL-lead were synthesised. Similar levels of stall were observed at each time point on the 8oxoG and Tg templates (Fig 6A, denaturing lanes 5–12), revealing 8oxoG is bypassed with similar efficiency to Tg by the eukaryotic replisome.

To analyse whether 8oxoG, like Tg, causes uncoupling of leading-strand synthesis before bypass, a pulse chase experiment in which Pol δ$^{cat}$ was added with the chase was performed on the UD and 8oxoG templates (Fig 6B). As was observed on the Tg template (Fig 2C), addition of Pol δ$^{cat}$ caused stall to persist across the time course on the 8oxoG template and no FL-lead was observed, whereas replication of UD was not affected by Pol δ$^{cat}$ (Fig 6B, lanes 10–12). This reveals that 8oxoG causes uncoupling of leading-strand synthesis when encountered by the replisome.

Since 8oxoG is tolerated by the replisome, but also causes transient uncoupling, we considered it likely that bypass was occurring via a leading-strand replicase switch from Pol ε to Pol δ, as is observed during Tg bypass. To test this hypothesis, we evaluated

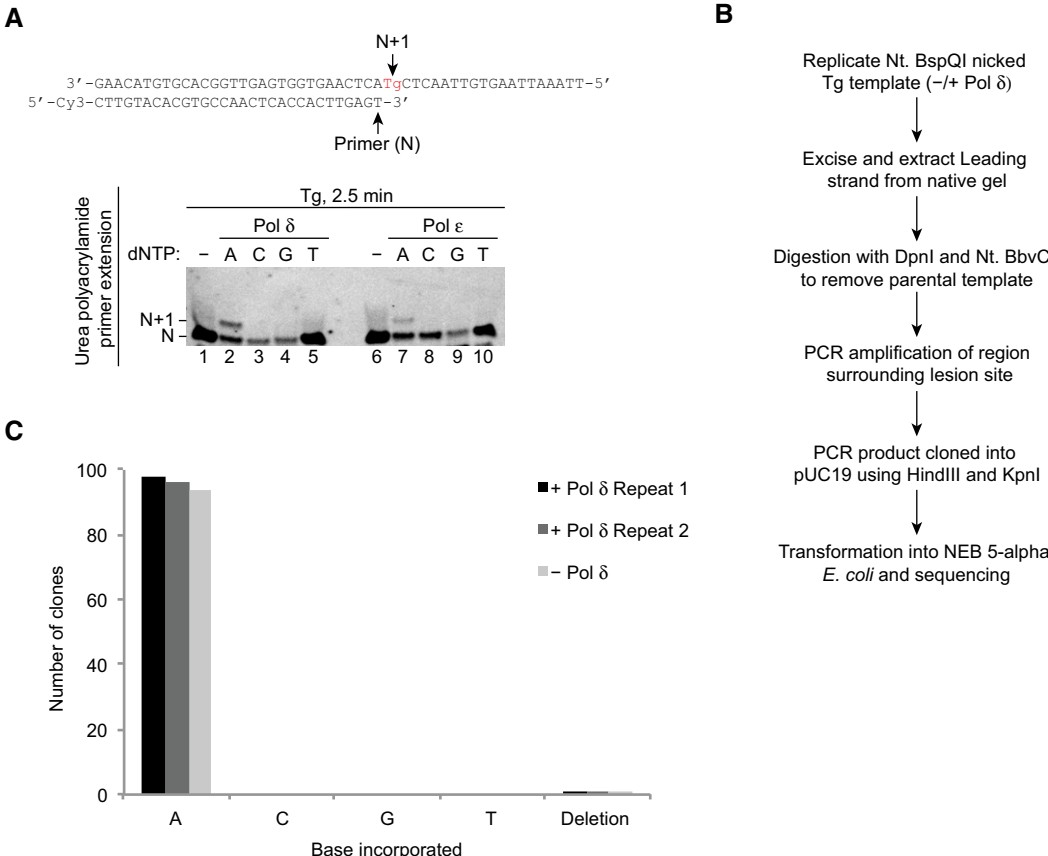

**Figure 5. Tg bypass in the absence and presence of Pol δ is error-free.**

A  Primer extension reactions on the Tg template with 5 nM Pol δ or Pol ε and either dATP, dCTP, dGTP or dTTP.

B  Outline of the method used to isolate and sequence replicated leading strands from reconstituted replication reactions.

C  Analysis of nucleotide incorporation opposite Tg from replication reactions processed as shown in (B). Three individual experiments were performed, two in the presence and one in the absence of Pol δ (10 nM), giving 99, 97 and 95 successful sequencing reads respectively.

**A**

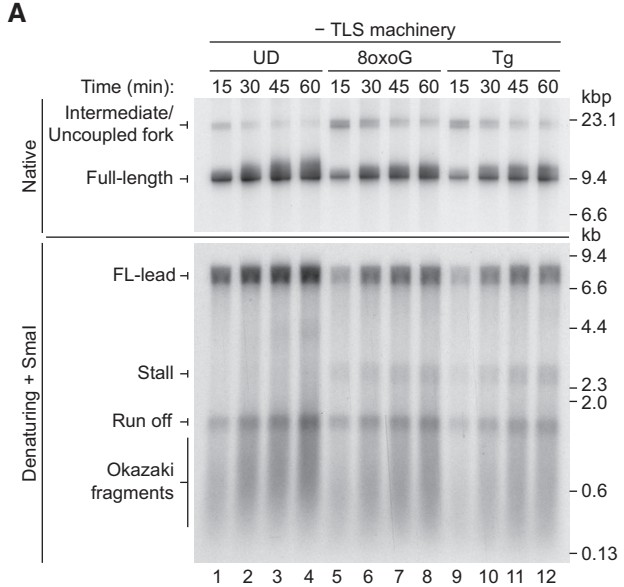

**B**

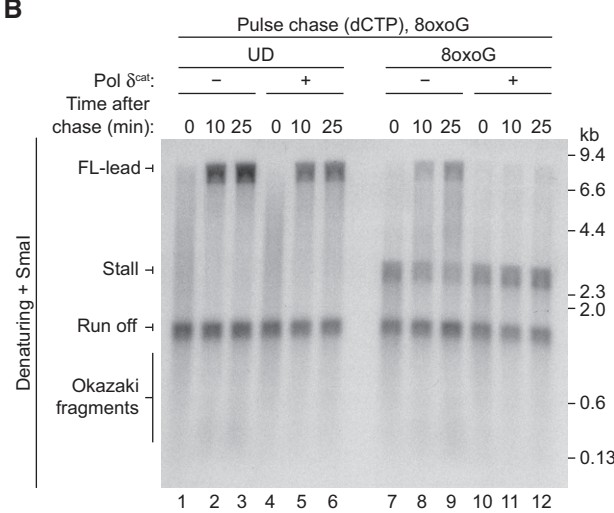

**C**

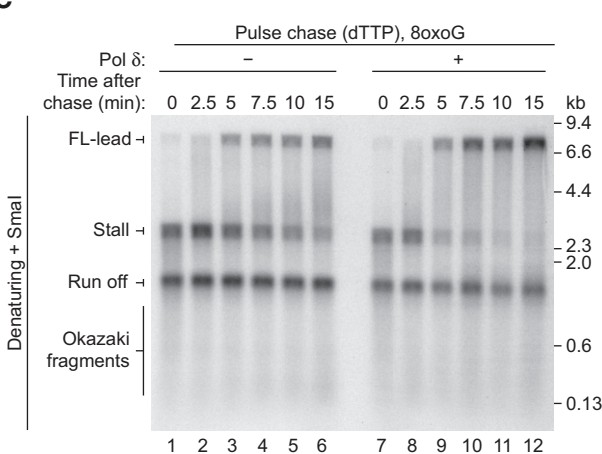

**Figure 6.   The replisome bypasses 8oxoG by a replicase switch mechanism.**

A    Standard replication assay on undamaged (UD), 8-oxoguanine (8oxoG), and thymine glycol (Tg) templates in the absence of TLS components.
B    Pulse chase reaction on UD and 8oxoG templates. Where present, Pol δ^cat (10 nM) was added directly after addition of the chase.
C    Pulse chase reactions on the 8oxoG template in the absence and presence of 10 nM Pol δ analysed on a denaturing gel.

the contribution of Pol δ to 8oxoG bypass in a pulse chase experiment where Pol δ was either added in the chase or omitted from the reaction (Fig 6C). Elevated dTTP was used in place of dCTP in the chase, as we considered that increased dCTP might stimulate 8oxoG bypass. In the absence of Pol δ, some FL-lead was produced; however, stall was also present across the time course (Fig 6C, lanes 1–6). In contrast, when Pol δ was present, stall was greatly depleted by the 5-min time point and a more intense FL-lead product was generated by the end of the reaction (Fig 6C, lanes 7–12). Together, these results reveal that, in addition to facilitating rapid and efficient Tg bypass, replicase switching can also promote tolerance of 8oxoG. This indicates that replicase switching may be a general mechanism for tolerating leading-strand base damage during replisome progression.

## Discussion

Our work has revealed that the yeast replisome is inherently tolerant of leading-strand thymine glycol and 8oxoG lesions, with damage bypass occurring via a replicase switch mechanism. When CMGE encounters the damaged base, the catalytic domain of Pol ε disengages from the nascent leading strand causing uncoupling of synthesis from template unwinding (Fig 7A). This promotes a switch from Pol ε to Pol δ at the 3′ end of the stalled leading strand (Fig 7B). Pol δ then promotes rapid and efficient lesion bypass, independently of canonical TLS factors, to facilitate the recoupling of leading-strand synthesis to CMGE (Fig 7C). Upon recoupling, a switch back to Pol ε occurs to allow rapid replication fork rates to resume (Fig 7D).

The replicase switch mechanism we describe is consistent with our prior assignment of Pol δ as a "first responder" to uncoupling of leading-strand synthesis (Guilliam & Yeeles, 2020a). Here, binding of Pol δ to the nascent leading strand limits the unregulated recruitment of error-prone TLS polymerases when uncoupling occurs but TLS polymerases are not required. We propose that replicase switching may enable a variety of leading-strand lesions, and potentially DNA secondary structures, to be bypassed without the deployment of DNA damage tolerance (DDT) pathways in any situation where Pol δ can bypass an obstacle more efficiently than Pol ε. When Pol δ alone cannot promote recoupling, such as at a CPD, DDT pathways are likely to be deployed (Guilliam & Yeeles, 2020a).

Previously, it was reported that Pol δ is unable to bypass Tg in primer extension assays and Pol ζ was determined to be responsible for TLS (Johnson *et al*, 2003). In contrast, we found that Pol δ can traverse Tg in primer extension reactions, with or without RFC/PCNA. These experiments were performed using conditions similar

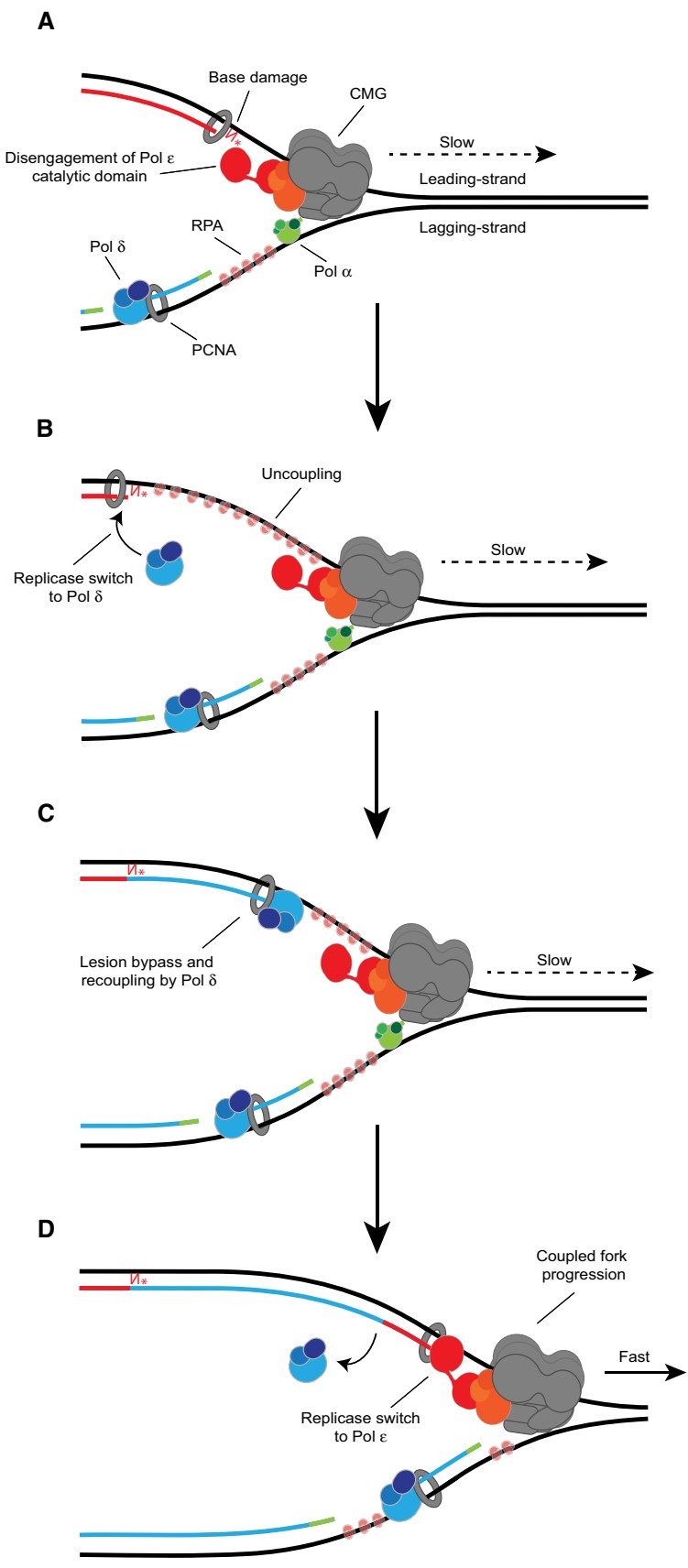

**Figure 7.**

◀

**Figure 7.  Model for the replicase switch bypass mechanism.**

A  Upon encounter with base damage in the template strand, the catalytic domain of Pol ε disengages from the nascent leading strand. Uncoupling of leading-strand synthesis from template unwinding slows replication fork rates. DNA strands are coloured based on the polymerase responsible for synthesis (Pol α: green, Pol δ: blue, Pol ε: red).

B  Uncoupled fork progression continues and a replicase switch from Pol ε to Pol δ occurs at the lesion.

C  Pol δ directly bypasses the damage independently of TLS components to promote recoupling.

D  A switch back to Pol ε allows rapid, coupled replisome progression to resume.

to those used for origin-dependent replication assays that support complete DNA replication at the *in vivo* rate (Yeeles *et al*, 2017) and at lower nucleotide concentrations than those used previously (Johnson *et al*, 2003). Tg bypass in primer extension assays did, however, appear less efficient than in origin-dependent replication assays. It is therefore possible that additional factors, sequence context (Clark & Beardsley, 1987) or fork architecture influence lesion bypass. *In vivo*, replicase-mediated lesion bypass might be enhanced at elevated nucleotide concentrations that occur in response to DNA damage (Chabes *et al*, 2003). Indeed, damage tolerance was significantly improved in a yeast strain lacking TLS polymerases when dNTP concentrations were increased (Sabouri *et al*, 2008).

Since Pol δ promotes efficient error-free lesion bypass of Tg at sub-nanomolar concentrations and physiological nucleotide levels, we suggest it would be favoured over Pol ζ, where possible, to reduce mutagenesis. However, if Pol δ cannot efficiently traverse the lesion, which may occur during some bypass events, TLS polymerases would be recruited by monoubiquitinated PCNA to prevent extended uncoupling. In agreement, Pol ζ and Rev1 have a reduced dependency on PCNA monoubiquitination when the interaction between Pol δ and PCNA is compromised (Tellier-Lebegue *et al*, 2017).

In higher eukaryotes, a number of TLS polymerases have been implicated in Tg bypass (Fischhaber *et al*, 2002; Kusumoto *et al*, 2002; Takata *et al*, 2006; Belousova *et al*, 2010; Yoon *et al*, 2010, 2014), many of which are absent in *S. cerevisiae*. Yeast Pol δ may therefore possess an increased ability to tolerate Tg due to the lack of these additional polymerases. However, studies of chicken DT40 cells suggest it can bypass some lesions at the replication fork in vertebrates (Hirota *et al*, 2015, 2016). Moreover, human Pol δ can bypass a number of single base lesions in primer extension assays, including 8oxoG (Choi *et al*, 2006; Meng *et al*, 2009; Schmitt *et al*, 2009), suggesting a role for Pol δ in replicase-mediated lesion bypass may be conserved in higher eukaryotes.

The results presented here further challenge the textbook view of Pol δ as simply the lagging-strand replicase, adding to the growing number of roles for the enzyme in leading-strand synthesis (Guilliam & Yeeles, 2020b). Furthermore, these findings directly demonstrate that the eukaryotic replisome has the inherent capacity to tolerate certain types of DNA damage, independently of canonical tolerance pathways, through a replicase switch mechanism.

# Materials and Methods

## Contact for reagent and resource sharing

Requests for further information and resources and reagents should be made to the Lead Contact, Joseph Yeeles (jyeeles@mrc-lmb.cam.ac.uk).

## Experimental model and subject details

Proteins were purified from *S. cerevisiae* W303 strains (genotype: MATa ade2-1 ura3-1 his3-11,15 trp1-1 leu2-3,112 can1-100 bar1::Hyg pep4::KanMX) harbouring integrated constructs for overexpression of the protein of interest. Synthetic gene constructs used for integration were codon optimised for high-level protein expression (Sharp & Li, 1987; Yeeles *et al*, 2015). Strain and plasmid details are given in the Appendix Tables S1–S3. Proteins were also purified from *Escherichia coli* RosettaTM 2(DE3) cells (Novagen) (genotype: F– ompT hsdSB(rB– mB–) gal dcm (DE3) pRARE2 (CamR) transformed with plasmids for overexpression of the desired protein.

## Method details

### Replication templates
The ZN3 ARS306 plasmid template used here for replication assays is identical to that used previously (Taylor & Yeeles, 2018). Tg and 8oxoG integration were performed in the same manner as described for CPD integration (see Appendix Tables S1 and S2 for oligonucleotide details) (Taylor & Yeeles, 2018). All plasmid templates were CsCl gradient purified and linearised with AhdI prior to replication, as previously described (Taylor & Yeeles, 2018).

A BspQI restriction site was generated in ZN3 by two rounds of site directed mutagenesis to form ZN3 Tg1. In the first round, an existing BspQI restriction site in ZN3 was removed (primer sequences in Appendix Table S2). In the second round, a new BspQI restriction site was added (primer sequences in Appendix Table S2). Integration of Tg and 8oxoG was performed in an identical manner to that described for ZN3 (Taylor & Yeeles, 2018). Following AhdI linearisation, the template was nicked with Nt. BspQI for 1 h at 37°C, before extraction of the DNA with phenol:chloroform:isoamyl alcohol 25:24:1 saturated with TE (Sigma-Aldrich P2069) and ethanol precipitation following standard practices. The DNA pellet was resuspended in TE buffer before use in replication assays.

### Yeast expression strain construction
The yTG2 and yTG7 yeast strains originating in this study (Appendix Table S3) were generated by transforming yJF1 (Frigola *et al*, 2013) with the respective linearised expression vectors (Appendix Table S4) following standard protocol. Codon optimised expression sequences (Sharp & Li, 1987; Yeeles *et al*, 2015) were synthesised using GeneArt Synthesis (Invitrogen) and cloned as stated in Appendix Table S3. yTG11 was generated by deletion of *REV3* in yAE34. To do this, a PCR product from pFA6-natMX6 (primer sequences in Appendix Table S2) was transformed into yAE34 to replace *REV3* with the natMX6 selectable marker. After selection, deletion of *REV3* was confirmed by PCR over 5′ and 3′ insertion sites. Details of the original expression vectors are given in

previous reports (Frigola *et al*, 2013; Coster *et al*, 2014). Additional strain and plasmid details can be found in Appendix Tables S3 and S4.

### Protein purification

With the exception of Rev1 and Pol ζ, all proteins were purified following the previously published protocols (Frigola *et al*, 2013; Coster *et al*, 2014; On *et al*, 2014; Yeeles *et al*, 2015, 2017; Devbhandari *et al*, 2017; Aria & Yeeles, 2019; Guilliam & Yeeles, 2020a). Pol ε$^{exo-}$ was purified using the method described for Pol ε mutants in (Aria & Yeeles, 2019). Individual purification strategies, affinity tags, final storage buffers and references to detailed purification protocols for each protein can be found in Appendix Table S5.

Rev1 and Pol ζ were both expressed in *S. cerevisiae*. Cells were incubated at 30°C in YEP + 2% raffinose until a density of 2–$3 \times 10^7$ cells per ml was reached. Protein expression was induced by addition of galactose to 2% and incubation for a further 3 h. Following harvesting, cells were resuspended in the buffers described below and frozen dropwise in liquid nitrogen. Lysis was performed using a liquid nitrogen-chilled pestle and mortar before storage at −80°C.

### Rev1 purification

Powder from 10 l of yTG2 was diluted 1:1 in Buffer A (40 mM Tris–HCl pH 7.5, 0.02% NP-40-S, 1 mM DTT, 10% glycerol) + 200 mM NaCl + 2 mM CaCl$_2$ + protease inhibitors. Cell lysate was clarified by centrifugation at 235,000 *g*, 45 min, 4°C and applied to 1 ml washed Calmodulin-Sepharose 4B (GE Healthcare) resin in a disposable gravity flow column (Bio-Rad). The column was washed with Buffer A + 200 mM NaCl + 2 mM CaCl$_2$ and eluted in Buffer A + 200 mM NaCl + 2 mM EDTA + 2 mM EGTA. The eluate was diluted with Buffer A to a final NaCl concentration of 150 mM and loaded onto a 1 ml HiTrap Heparin column equilibrated in Buffer A + 150 mM NaCl + 0.5 mM EDTA. The column was washed using the same buffer and proteins were eluted using a 30 ml gradient to 1 M NaCl. Peak fractions containing purified Rev1 were pooled before dialysis overnight against Buffer B (25 mM HEPES-KOH pH 7.6, 10% glycerol, 1 mM DTT, 0.02% NP-40-S) + 300 mM KOAc + 0.5 mM EDTA. The dialysate was retrieved, aliquoted, frozen in liquid nitrogen and stored at −80°C.

### Pol ζ purification

Powder from 10 l of yTG7 was diluted 1:1 in Buffer C (50 mM HEPES-KOH, 500 mM KOAc, 10% glycerol, 1 mM DTT, 0.02% NP-40-S, 0.5 mM EDTA) + protease inhibitors. Following clarification of the lysate by centrifugation (235,000 *g*, 45 min, 4°C), 2 ml Anti-FLAG M2 affinity gel (Sigma-Aldrich) was added to the soluble extract and incubated with gentle mixing at 4°C for 1 h. The resin was collected in a disposable gravity flow column (Bio-Rad) and washed in Buffer C. Further washes were performed with Buffer B + 200 mM KOAc + 5 mM Mg(OAc)$_2$ + 0.5 mM ATP and Buffer B + 200 mM KOAc. Proteins were eluted by incubation of the resin in Buffer B + 200 mM KOAc + 0.25 mg/ml 3xFLAG peptide for 10 min on ice. The eluate was supplemented with 2 mM CaCl$_2$ and applied to 1 ml Calmodulin-Sepharose 4B (GE Healthcare) resin in a disposable gravity flow column (Bio-Rad). The resin was washed with Buffer B + 200 mM KOAc + 2 mM CaCl$_2$, before elution with Buffer B + 200 mM KOAc + 2 mM EDTA + 2 mM EGTA. The eluate

was bound to a 1 ml HiTrap Heparin column equilibrated in Buffer B + 200 mM KOAc. Following washing, a 30 ml gradient elution to 1.5 M KOAc was performed. Peak fractions were pooled and concentrated before being aliquoted, frozen in liquid nitrogen and stored at −80°C.

### Standard replication assays

Replication reactions were performed as described previously (Taylor & Yeeles, 2018; Guilliam & Yeeles, 2020a). 5 nM AhdI linearised ZN3, or ZN3 Tg1, was incubated at 24°C for 10 min with 75 nM Cdt1-Mcm2-7, 45 nM Cdc6, 20 nM ORC and 50 nM DDK, in buffer containing 25 mM HEPES-KOH (pH 7.6), 100 mM potassium glutamate, 10 mM Mg(OAc)$_2$, 0.02% NP-40-S, 5 mM ATP and 0.1 mg/ml BSA, to promote MCM loading and phosphorylation. S-CDK was added to a final concentration of 150 nM and the reaction was incubated for an additional 5 min, before being diluted 4-fold with replication buffer to give the final concentrations: 25 mM HEPES-KOH (pH 7.6), 250 mM potassium glutamate, 0.02% NP-40-S, 10 mM Mg(OAc)$_2$, 0.1 mg/ml BSA, 5 mM ATP, 200 μM CTP, 200 μM GTP, 200 μM UTP, 30 μM dATP, 30 μM dCTP, 30 μM dGTP, 30 μM dTTP, and 1 μCi [α-$^{32}$P]-dCTP (or [α-$^{32}$P]-dTTP). DNA synthesis was initiated by incubation at 30°C with the following proteins (final concentrations): 30 nM Dpb11, 100 nM GINS, 40 nM Cdc45, 10 nM Pol ε, 5 nM MCM10, 20 nM Ctf4, 100 nM RPA, 20 nM Csm3/Tof1, 10 nM Mrc1, 20 nM RFC, 20 nM PCNA, 10 nM TopoI, 20 nM Pol α, 20 nM Sld3/7 and 20 nM Sld2. Except where otherwise stated in figures or figure legends, Pol δ was used at a final concentration of 10 nM when present in replication reactions. TLS machinery components were used at the following final concentrations: 10 nM Pol η, 5 nM Pol ζ, 5 nM Rev1, 200 nM Rad6–Rad18, 25 nM Uba1 and 1 μM ubiquitin. In oligonucleotide-mediated replication restart experiments, the RPA concentration was lowered to 60 nM and the oligonucleotides were used at final concentration of 100 nM.

### Pulse chase experiments

Pulse chase experiments were performed the same as standard reactions except that the concentration of dCTP (or dTTP) was lowered to 5 μM for the 5-min pulse phase and elevated to 600 μM for the chase phase.

### Post-reaction sample processing

Reactions were stopped by addition of EDTA to 25 mM before deproteinisation with proteinase K (8 U/ml, NEB)—SDS (0.1%) at 37°C for 15 min. Reaction products were extracted using phenol–chloroform–isoamyl alcohol (Sigma-Aldrich), and unincorporated nucleotide was removed with illustra G-50 columns (GE Healthcare) following the manufacturers instructions. Prior to denaturing agarose gel analysis, samples were digested with SmaI to reduce heterogeneity in product sizes as described previously (Taylor & Yeeles, 2018). Denaturing and native agarose gel electrophoresis was performed as previously detailed (Taylor & Yeeles, 2018). Before resolution on 6% polyacrylamide (Bis-Acrylamide 19:1—Fisher Scientific), 7 M urea denaturing gels, replication reaction products were digested with SwaI (NEB) and BamHI-HF (NEB), and electrophoresis was performed as described previously (Guilliam & Yeeles, 2020a).

After running, polyacrylamide and native agarose gels were immediately dried onto 3MM chromatography paper (GE Healthcare). Denaturing agarose gels were fixed with 5% trichloroacetic acid solution as detailed previously (Guilliam & Yeeles, 2020a) before drying. For visualisation, gels were exposed on BAS-IP MS Storage Phosphor Screens (GE Healthcare) and scanned on a Typhoon phosphorimager (GE Healthcare). Agarose gels were additionally autoradiographed using Amersham Hyperfilm MP (GE Healthcare) for presentation.

### Generation of sequencing ladder

The sequencing ladder used to map the 3′-end of the stalled nascent leading strand was generated using a USB Sequenase kit (Affymetrix) following the manufacturers instructions. A primer (sequence given in Appendix Table S2) was annealed to the UD ZN3 template, so that the 5′-end mapped to the SwaI restriction site and extended to generate the ladder.

### Isolation of replicated leading strands for sequencing

Replication reactions were performed without [$\alpha$-$^{32}$P]-dCTP on Tg-containing Nt. BspQI nicked ZN3 Tg1 for 30 min in the absence or presence of 10 nM Pol δ. Reactions were quenched with EDTA, and deproteinisation and phenol-chloroform extraction were performed, as for standard replication reactions. Products were digested with SmaI (NEB) for 30 min at 25°C before quenching by addition of EDTA to 25 mM and resolution on a 0.8% native agarose gel, run overnight at 20 V. A molecular weight marker corresponding to the size of the expected duplex leading-strand replication product was run alongside to assist with band extraction. Gels were stained with SYBR Safe (Invitrogen) diluted 1:10,000 in TAE for 20 min, before visualisation on a blue-light transilluminator. Bands corresponding to the expected location of the replicated leading-strand products were excised and gel extraction was performed using a Qiagen Gel Extraction Kit, following the manufacturers instructions. The extracted DNA was digested with Nt. BbvCI (NEB) and DpnI (NEB) for 90 min at 37°C before heat inactivation at 80°C for 20 min. The region surrounding the lesion was PCR-amplified (primer sequences in Appendix Table S2, Byp_FID_FWD and REV) and cloned into pUC19 using HindIII-HF (NEB) and KpnI-HF (NEB). Following transformation into NEB 5-alpha Competent *E. coli*, individual clones were selected and grown overnight in a 96-well plate before Sanger sequencing with the M13F primer (Source BioScience) using the Source BioScience Bugs2Bases service.

### Primer extension assays

Primer extension was monitored on synthetic primer–template oligonucleotides (ATDBio) where the 5′-end of the primer was fluorescently labelled with Cy3 (sequences available in Appendix Table S2). Oligonucleotides were annealed by heating to 95°C in TE buffer for 3 min before gradual cooling and storage at −20°C until use. Primer–template DNA (10 nM) was pre-incubated with 5 nM RFC and 20 nM PCNA (when present) at 30°C for 1 min in buffer containing 25 mM HEPES-KOH (pH 7.6), 100 mM potassium glutamate, 10 mM Mg(OAc)$_2$, 0.1 mg/ml BSA, 1 mM ATP, 30 μM dATP, 30 μM dCTP, 30 μM dGTP and 30 μM dTTP. DNA synthesis was initiated by addition of Pol δ or Pol ε at the concentrations indicated in individual figures or legends and reactions were further incubated at 30°C for 10 min. For analysis of nucleotide incorporation

opposite Tg in Fig 5A, reactions contained 250 μM of the individual dNTP being tested and were incubated for 2.5 min before quenching. Quenching was performed by addition of 1 volume of loading buffer (95% formamide, 0.05% xylene cyanol, 0.05% bromophenol blue, 20 mM EDTA and 200 nM competitor DNA (sequence in Appendix Table S2). Samples were heated to 95°C for 2 min and resolved on 15% polyacrylamide (Bis-Acrylamide 19:1—Fisher Scientific) 7 M urea denaturing gels. After running, gels were immediately scanned using a Typhoon imager (GE Healthcare) to detect Cy3-labelled extension products.

### Quantification and statistical analysis

Quantification was performed using ImageJ software. The percentage of bypass in Fig 3F was quantified by generating profiles of each gel lane in ImageJ. A straight line was manually fit to the background baseline, and stall and bypass bands were quantified. Bypass percentage was calculated using the following equation: % bypass = (bypass/(bypass + stall)) × 100.

## Data availability

This study includes no data deposited in external repositories.

**Expanded View** for this article is available online.

## Acknowledgements

This work was supported by the Medical Research Council, as part of United Kingdom Research and Innovation (MRC grant No. MC_UP_1201/12 to J.T.P.Y). T.A.G. is supported by a Sir Henry Wellcome Postdoctoral Fellowship from the Wellcome Trust (213596/Z/18/Z).

## Author contributions

TAG performed the experiments. TAG and JTPY wrote the manuscript.

## Conflict of interest

The authors declare that they have no conflict of interest.

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
