## [Review Process File · The EMBO Journal]

The eukaryotic replisome tolerates leading-strand base damage by replicase switching

Thomas Guilliam and Joseph Yeeles

DOI: [10.15252/embj.2020107037](https://doi.org/10.15252/embj.2020107037)

Corresponding author(s): Joseph Yeeles (jyeeles@mrc-lmb.cam.ac.uk)

Review Timeline:

Submission Date:	14th Oct 20
Editorial Decision:	2nd Nov 20
Revision Received:	9th Dec 20
Accepted:	7th Jan 21

Editor: Hartmut Vodermaier

Transaction Report:

Thank you for submitting your manuscript on Pol delta bypass of leading strand base damage to The EMBO Journal. Three expert referees have now reviewed the study, and in light of their overall positive comments, we would be happy to consider a revised version further for publication in our journal. As you will see from the reports copied below, referees 1 and 2 raise a number of specific issues that would need to be addressed prior to acceptance, with clarification of the technical queries (e.g. ref 1 #1) being particularly important. Please note that it is our policy to allow only a single round of major revision, making it important to carefully answer to all referee points at this stage.

Since I am aware of the difficulties regarding lab access and experimental work in the present COVID-19 pandemic situation, I would be available for discussing options for how and within what timeline to best revise this study at any time, so please do not hesitate to contact me in case you should have any questions/comments regarding the reviews. Our 'scooping protection' (meaning that competing work appearing elsewhere in the meantime will not affect our considerations of your study) obviously remains valid also during a potential extension of the revision period.

Referee #1:

Guilliam and Yeeles recently published a paper where they studied the mechanism by which the replisome by-pass a cyclobutene pyrimidine dimer (CPD). Here they asked how an origin dependent replication fork handle two different base-damages, Thymine glycol (Tg) and 8-oxoG. Based on an earlier study suggesting that Pol Delta is unable to bypass a Tg they presumed that that Pol Zeta and Rev1 would be required to bypass the Tg. Surprisingly, they found that Pol Zeta could be omitted and that a polymerase switch allowed Pol Delta to bypass the Tg after Pol Epsilon stalled at the damaged base. Furthermore, the results imply that Pol Epsilon resumes leading strand synthesis within less than 265 nt from the lesion. A similar switch was shown when the replisome encountered an 8-oxoG lesion in the leading strand template. The authors propose that the observed switch mechanism where Pol Delta participate sin leading strand synthesis across lesions

can be generalized when lesions are present on the leading strand. This is a well-designed and clearly presented manuscript with novel findings. However, there are a few things that need a clarification.

Major points

1. Pol32 and Pol31 are two non-catalytic subunits that are shared by both Pol delta and Pol Zeta. The description of how Pol Delta and Pol Zeta is purified reveals that both Pol delta and Pol Zeta are affinity-purified with help of a CBP-tag on the C-terminus of Pol32 followed by a HiTrap Heparin HP. Could you please clarify how you can be certain that Pol Zeta is not present as a contamination in the purified Pol delta. It would strengthen the paper if that was clarified in the manuscript since your results are very different compared to what was reported in Johnson et al 2003. Just a small amount of Pol Zeta (below detection level on the SDS-PAGE) could have sufficient activity to support bypass-synthesis.

2. Please quantify products and calculate the bypass efficiency when comparing reactions in the absence and presence of the TLS machinery.

3. Fig1C, Could you please comment on why a small fraction of the lesion in the template appears to be impassible.

4. Page 7, line 4-7. The scrambled oligonucleotide is supposed to not promote replication restart. Despite that is a product observed in Fig 2B, lane 4. Please comment on how the replication restart product can be observed in lane 4. Please quantify the replication restart products and full-length products to estimate replication restart efficiency in each lane, lane 4-6.

5. Considering that there is ssDNA available for replication restart, it is surprising that Pol alfa is unable to synthesize a primer. Could you please comment on why that is the case?

6. Page 11, lines 9-12, "Pol δ stimulated conversion of stall to bypass across the titration range. Quantification revealed that 0.63 nM Pol δ was sufficient to substantially enhance Tg bypass, with only a slight further increase at higher concentrations (Fig 3F). This demonstrates that sub-nanomolar concentrations of Pol δ promote rapid and efficient leading-strand Tg bypass." What is the actual concentration of active forks? The methods section explains that the reaction is initiated with 5 nM plasmid, but only a small fraction of the origins are fired. Thus, the actual concentration of active replication forks could be much lower than 0,63 nM, and that would result in a molar excess of Pol Delta over active forks. Overall, this may influence the efficiency by which Pol delta bypass lesions on the leading strand.

7. Page 13, Fig 4B, could you please comment on the product seen at position +24 that is only visible in the presence of Tg, and not CPD containing substrate.

8. Fig 3G and EV2, the primer is too short for Pol epsilon to be fully engaged in processive DNA synthesis. Thus, the bypass efficiency may or may not be affected if a 30 nt long primer is used instead. Furthermore, please include a control where RFC/PCNA is omitted from the Pol epsilon reaction to clarify whether PCNA stimulates Pol epsilon under the conditions used in the primer-extension assay. Pol delta is shown in EV2C, but not Pol epsilon.

9. Page 18, lines 19-21. The authors refer to papers suggesting that increased dNTP pools may increase the efficiency by which DNA lesions are bypassed. The increased dNTP pools may also

affect the balance between Pol epsilon and Pol delta DNA synthesis during bypass of lesions on the leading strand. Having said that, there is no contradiction between the findings in this paper and the paper by Sabouri et al 2008. However, it would be good if the authors could include one or two assays exploring how an increased dNTP concentration could influence the switch to Pol Delta during leading strand synthesis, considering that this would influence the main message of the manuscript.

Minor points

1. Page 2, line 15, "We propose that replicase switching promotes continued leading-strand synthesis whenever the replisome encounters leading-strand damage that is bypassed more efficiently by Pol δ than by Pol ϵ .". The last sentence of the abstract is a strong generalization considering the large variety of DNA lesions found in vivo and many factors that can influence the activity of Pol Delta and Pol Epsilon. For example, the authors mention the dNTP concentrations which has not really been investigated in this manuscript.
2. Page 17, line 5-7, please rephrase the wording "is likely to" with "may". So far, the studies of three DNA lesions are pointing in that direction but I believe that it is too early to generalize considering the diversity of DNA lesions and also other effectors such as e.g. dNTP concentrations., that may influence the process.

Referee #2:

The paper by Guilliam and Yeeles, entitled, "The eukaryotic replisome tolerates leading-strand base damage by replicase switching", reports on a study of the bypass of leading strand lesions, Tg and 8-oxoG, by the yeast replisome. The authors use a yeast replisome reconstituted with purified proteins, as well as purified pols ϵ and δ to investigate leading-strand lesion bypass. The authors observe that the replisome is tolerant of the Tg and 8oxo-G lesions, and that bypass of these lesions is not dependent on the specialized TLS polymerases, but rather majority of bypass depends on the lagging strand replicase, pol δ . Based on the results the authors conclude that when pol ϵ encounters the lesion it uncouples from the helicase, which permits pol δ to bind, incorporate the correct nucleotide opposite the lesion, and continue synthesis until it catches up with the helicase. At this point a switch to pol ϵ , and its recoupling with the helicase occurs to complete leading strand synthesis.

The results support the authors' conclusions, they are well presented and discussed, and the reasoning is clear. Overall it's a very nice study and a well written paper. It will be of interest to a wide readership of the EMBO Journal.

The authors show that pol ϵ is also able to bypass a template Tg lesion, however bypass by pol δ is more rapid and more efficient than by pol ϵ it is not clear why this is the case. Could the exonuclease of pol ϵ be responsible for the less efficient bypass of the Tg lesion, causing idling of the polymerase at the site of the lesion and resulting in its uncoupling from the helicase? Could the authors comment? It would be interesting to perform bypass reactions with an exonuclease deficient variant of pol ϵ . Having said this, I do not suggest that this experiment is necessary for this paper.

Few additional minor points

Figure 2B, lane 4, there are clear bands at the "stall" and "oligo restart " positions-products of Tg

bypass synthesis in the presence of the scrambled oligonucleotide S, why is that? The authors should comment on this.

Legend to Figure 2A, for clarity, the schematic showing the replication products should be described in more detail; what do the green sections and the dashed line in the synthesized strand represent?

Page 8, line 10, "On the UD template, FL-lead was generated both in the absence and presence of Pol δ cat (Fig 2C, lanes 1-6), confirming the mutant does not inhibit coupled leading strand synthesis." However, inspection of Fig 2C suggests that there is less FL-lead product with the UD template in the presence of pol δ cat (compare lanes 1,2,3 with 4,5,6). The same appears to be true in the experiment presented in Fig 6B, again compare lanes 1,2,3, with lanes 4,5,6. Did the authors quantitate the amount of FL-lead product in these reactions? Is my assessment accurate? If so, please comment why pol δ cat would have an effect in this case.

Figure 3C, shouldn't the lesion in the template strand be Tg not CPD?

Referee #3:

This paper investigates the consequence of budding yeast replisome encounters with damaged DNA bases (thymine glycol [Tg], 8-oxo-G [8oG]) on the the leading strand template. Using the reconstituted budding yeast DNA replication system the authors demonstrate that yeast replisomes are inherently able to bypass Tg or 8oG on the leading strand independent from TLS polymerases. Experimental evidence demonstrates that leading strand synthesis by Pol-epsilon is transiently uncoupled from replisome progression at the site of DNA damage, while re-coupling is mediated after error-free bypass by Pol-delta. The findings highlight the distinct impact of Tg and 8oG on replisome progression from that of CPD damage, as the latter has been demonstrated by the authors previously to be dependent TLS. The data uncover that transient switching between Pol-epsilon and Pol-delta promotes inherent error-free bypass of Tg and 8oG damage sites, thus revealing a further role for the lagging strand polymerase, Pol-delta, in leading strand synthesis.

The data is robust and of high quality, the elegant assays have been previously established by the authors for the study of replisome collisions with CPD damage (Guillian & Yeeles, NSMB, 2020). I recommend publication of this manuscript in its current form.

Dear Editors and Reviewers,

Thank you for the time you have spent reviewing our manuscript. We are confident that we have addressed all the specific points raised by the three reviewers. Our responses to the individual comments are listed below.

Referee #1:

Guilliam and Yeeles recently published a paper where they studied the mechanism by which the replisome by-pass a cyclobutene pyrimidine dimer (CPD). Here they asked how an origin dependent replication fork handle two different base-damages, Thymine glycol (Tg) and 8-oxoG. Based on an earlier study suggesting that Pol Delta is unable to bypass a Tg they presumed that that Pol Zeta and Rev1 would be required to bypass the Tg. Surprisingly, they found that Pol Zeta could be omitted and that a polymerase switch allowed Pol Delta to bypass the Tg after Pol Epsilon stalled at the damaged base. Furthermore, the results imply that Pol Epsilon resumes leading strand synthesis within less than 265 nt from the lesion. A similar switch was shown when the replisome encountered an 8-oxoG lesion in the leading strand template. The authors propose that the observed switch mechanism where Pol Delta participate in leading strand synthesis across lesions can be generalized when lesions are present on the leading strand. This is a well-designed and clearly presented manuscript with novel findings. However, there are a few things that need a clarification.

Major points

1. Pol32 and Pol31 are two non-catalytic subunits that are shared by both Pol delta and Pol Zeta. The description of how Pol Delta and Pol Zeta is purified reveals that both Pol delta and Pol Zeta are affinity-purified with help of a CBP-tag on the C-terminus of Pol32 followed by a HiTrap Heparin HP. Could you please clarify how you can be certain that Pol Zeta is not present as a contamination in the purified Pol delta. It would strengthen the paper if that was clarified in the manuscript since your results are very different compared to what was reported in Johnson et al 2003. Just a small amount of Pol Zeta (below detection level on the SDS-PAGE) could have sufficient activity to support bypass-synthesis.

To exclude the possibility that contaminating Pol ζ is responsible for the lesion bypass we observe with Pol δ we have deleted *REV3* in the Pol δ overexpression strain. Addition of Pol δ purified from the *REV3* deletion strain stimulates leading-strand Tg bypass in the replication assay (new Fig EV2B), as seen upon Pol δ addition previously. We can therefore rule out the possibility that the bypass we observe is due to contamination with Pol ζ .

2. Please quantify products and calculate the bypass efficiency when comparing reactions in the absence and presence of the TLS machinery.

It is important to emphasise that the key result from these experiments is that no canonical TLS factors are required for leading-strand Tg bypass by the replisome. This is clear from inspection of the gels and comparison to the CPD template reactions where TLS factors are required. We are not making the case that TLS factors don't additionally support a subset of bypass events. We therefore do not see the value of quantification of any minor differences in bypass efficiency for these experiments. Moreover, we feel that including quantification would distract from the central message that the TLS machinery is not required for Tg bypass by the replisome.

In addition, it is difficult to accurately and reliably quantify subtle changes in bypass efficiency from these gels due to the high amount of background. We therefore could not be confident in quoting specific values for bypass efficiency. For these reasons we believe that it is not appropriate or necessary to provide quantification for these experiments. We acknowledge that in the original manuscript we stated that "Tg bypass and the completion of leading-strand replication occurred equally efficiently in the presence and absence of the TLS machinery". We have therefore modified this statement to "Tg bypass and the completion of leading-strand replication occurred **with similar efficiency** in the presence and absence of the TLS machinery".

3. Fig1C, Could you please comment on why a small fraction of the lesion in the template appears to be impassible.

The small amount of stall product still present by the final 45 min time point may be due to a number of reasons, but importantly these do not affect the overall result or interpretation of the experiment, i.e., that a leading-strand thymine glycol is bypassed by the replisome. Firstly, this experiment is not under pulse chase conditions. This means that origins will continue to fire over the time course and therefore replication forks will continually encounter the lesion and generate stall products before bypass occurs. Secondly, it is possible that a small minority of the template contains a nick close to the lesion site since we nick the leading strand before ligating in the thymine glycol oligo when constructing the template (see methods). Third, a small amount of the template may contain a single-strand break caused by instability at the lesion site. We previously observed that an abasic site or tetrahydrofuran lesion at the same location could cause breakage of the template (Taylor and Yeeles, 2019, *J. Mol. Biol.*).

4. Page 7, line 4-7. The scrambled oligonucleotide is supposed to not promote replication restart. Despite that is a product observed in Fig 2B, lane 4. Please comment on how the replication restart product can be observed in lane 4. Please quantify the replication restart products and full-length products to estimate replication restart efficiency in each lane, lane 4-6.

We have used the same scrambled oligonucleotide in previous publications (Guilliam and Yeeles, 2020, *Nat. Struct. Mol. Biol.*; Taylor and Yeeles, 2018, *Mol. Cell*) on a CPD template, which also causes uncoupling, and do not detect this faint reaction product. Moreover, the same product can be seen in Figure 1C lane 6 and Figure 1D lane 6 which do not contain the scrambled oligonucleotide. This faint band is therefore not due to the presence of the oligonucleotide but is specific to the Tg template. The product is therefore likely to be caused by a nick left over from template generation or a break at the lesion site, with subsequent labelling of the 3'-end of the downstream ~ 5.2 kb template fragment during the course of the reaction. Unfortunately, this fragment is of roughly the same size as the restart product from the 21 nt re-priming oligonucleotide. However, importantly, the intensity of this end-labelled product is much less than that of the oligonucleotide-restart products and does not affect the overall interpretation of the key result i.e., that the thymine glycol lesion causes uncoupling of leading-strand synthesis from template unwinding. Moreover, we confirm in Figure 2C using a different approach that uncoupling is occurring.

We do not believe that it is possible to accurately quantify replication restart efficiency in these reactions, or that it would be useful for interpretation, due to the high level of background and varying efficiency of the reactions. We therefore do not want to quote specific numbers for restart efficiency. Instead, we have included lane profiles for lanes 4-6 which are normalised against the run-off product (new Figure 2C). Here, it is clear that addition of the 21 nt oligo causes a decrease in FL-lead production and a concomitant appearance of the more intense discontinuous restart product. In comparison, the 265 nt oligo causes a lesser decrease in FL-lead and a less intense restart product. This confirms that the 265 nt oligo competes less well with direct lesion bypass compared to the 21 nt oligo, and therefore in some instances lesion bypass and recoupling has taken place before 265 nt of uncoupling has occurred. Moreover, these lane profiles reveal that the end-labelled product in lane 4 is of a much lower intensity than either of the restart products in lanes 5 and 6.

5. Considering that there is ssDNA available for replication restart, it is surprising that Pol alpha is unable to synthesize a primer. Could you please comment on why that is the case?

We have previously demonstrated that leading-strand repriming in the reconstituted replication system is extremely inefficient due to inhibition of Pol α by RPA (Taylor and Yeeles, 2018, *Mol. Cell*). We suspect that the reason Pol α can efficiently prime the lagging- but not leading-strand beyond damage will relate to how Pol α is functionally targeted to replication forks for priming. However, at present, it is not known how this occurs.

6. Page 11, lines 9-12, "Pol δ stimulated conversion of stall to bypass across the titration range. Quantification revealed that 0.63 nM Pol δ was sufficient to substantially enhance Tg bypass, with only a slight further increase at higher concentrations (Fig 3F). This demonstrates that sub-nanomolar concentrations of Pol δ promote rapid and efficient leading-strand Tg bypass. "

What is the actual concentration of active forks? The methods section explains that the reaction is initiated with 5 nM plasmid, but only a small fraction of the origins are fired. Thus, the actual concentration of active replication forks could be much lower than 0,63 nM, and that would result in a molar excess of Pol Delta over active forks. Overall, this may influence the efficiency by which Pol delta bypass lesions on the

leading strand.

The referee is correct that not all origins fire during replication. However, the key point of this experiment is that very low concentrations of Pol δ (0.63 nM) can promote efficient Tg bypass on the leading strand. Importantly, this concentration of Pol δ is much lower than either of the other replicative polymerases, Pol ϵ (10 nM) or Pol α (20 nM), and much lower than the concentration of Pol δ typically used in this system in previous publications (5-10 nM). Similarly, it is significantly lower than the concentration of any other protein in the replication assay. The fact that Pol δ can outcompete the other polymerases for access to the stalled nascent leading strand and promote efficient Tg bypass despite being present at 15-30 fold lower concentration suggests that it would also be favoured over the other polymerases for bypass *in vivo*.

It is not technically possible for us calculate the actual concentration of active forks in the replication assay. Moreover, this is complicated by the fact that Pol δ will also be titrated away onto Okazaki fragments on the lagging strand from both the leftward and rightward moving forks. Since Fen1 and ligase are not present in these experiments, fragments remain unligated. Therefore, Pol δ may remain bound to unligated Okazaki fragments, the number of which will increase over the course of the reaction. As such, even if the concentration of active forks were less than 0.63 nM it would not necessarily mean that Pol δ would be present at an excess over free 3'-ends. Regardless, our statement that “sub-nanomolar concentrations of Pol δ promote rapid and efficient leading-strand Tg bypass”, remains technically correct despite this.

7. Page 13, Fig 4B, could you please comment on the product seen at position +24 that is only visible in the presence of Tg, and not CPD containing substrate.

In this experiment we cleaved reactions products with Swal which cuts upstream of the lesion and BamHI which cuts downstream (Figure 3C). BamHI will therefore only cleave reaction products if lesion bypass and extension past the BamHI site occurs. Stalling at the lesion will generate a 165 nt product defined by cleavage at the 5'-end by Swal and stalling at the 3'-end. Bypass will produce a 187 nt product defined by cleavage at the 5'-end by Swal and cleavage at the 3'-end by BamHI. The larger (187 nt) product seen only on the Tg template is due to bypass of the lesion and

cleavage by BamHI. A more intense product is seen in the presence of Pol δ due to the increased bypass efficiency in its presence (lane 6). Since Pol η is required for CPD bypass (and is not present in this experiment) (Guilliam and Yeeles, 2020, *Nat. Struct. Mol. Biol.*), no 187 nt product is generated on the CPD template.

8. Fig 3G and EV2, the primer is too short for Pol epsilon to be fully engaged in processive DNA synthesis. Thus, the bypass efficiency may or may not be affected if a 30 nt long primer is used instead. Furthermore, please include a control where RFC/PCNA is omitted from the Pol epsilon reaction to clarify whether PCNA stimulates Pol epsilon under the conditions used in the primer-extension assay. Pol delta is shown in EV2C, but not Pol epsilon.

We have now performed primer extension assays to compare Tg bypass efficiency by Pol ϵ using a 20 nt or 30 nt long primer. We do not detect any significant difference in bypass efficiency when the 30 nt primer is used instead of the 20 nt primer (new Fig EV3D).

We have now included the requested control experiment (new Fig EV3E) to confirm that RFC/PCNA stimulates Pol ϵ activity in the primer extension assay.

9. Page 18, lines 19-21. The authors refer to papers suggesting that increased dNTP pools may increase the efficiency by which DNA lesions are bypassed. The increased dNTP pools may also affect the balance between Pol epsilon and Pol delta DNA synthesis during bypass of lesions on the leading strand. Having said that, there is no contradiction between the findings in this paper and the paper by Sabouri et al 2008. However, it would be good if the authors could include one or two assays exploring how an increased dNTP concentration could influence the switch to Pol Delta during leading strand synthesis, considering that this would influence the main message of the manuscript.

To investigate the effect of dNTP concentrations on Tg bypass and the switch to Pol δ on the leading strand, we have performed a pulse chase experiment at 30 μ M and 150 μ M dNTPs with Pol δ either added or omitted from the chase (new Fig EV2C). At 30 μ M dNTPs, addition of Pol δ in the chase (\sim 3.5 min into the reaction) stimulated the production of FL-lead and decreased the amount of stall, as observed previously

(compare lanes 2 and 3 to 4 and 6). At 150 μM dNTPs, Tg bypass in the absence of Pol δ was more efficient than at 30 μM dNTPs, as shown by the increased amount of FL-lead and decreased stall product (compare lanes 1-3 to 7-9). Despite this enhanced lesion bypass efficiency in the absence of Pol δ at higher dNTP concentrations, addition of Pol δ in the chase still stimulated the generation of FL-lead under these conditions (compare lanes 8 and 9 to 11 and 12). This suggests that a switch to Pol δ still occurs during leading-strand Tg bypass at 150 μM dNTPs, at least in a subset of lesion bypass events.

The enhanced lesion bypass observed at 150 μM dNTPs in the absence of Pol δ could be due to bypass by CMG-bound Pol ϵ , thereby preventing uncoupling. Alternatively, the increased dNTP concentration may stimulate Tg bypass by free Pol ϵ following uncoupling. To distinguish between these two possibilities and investigate whether uncoupling still occurs at 150 μM dNTPs, we repeated the same assay but either added or omitted Pol δ^{cat} in the chase (new Fig EV2D). Here, at both 30 μM and 150 μM dNTPs, addition of Pol δ^{cat} in the chase inhibited the generation of FL-lead. This confirms that even at high dNTP concentrations a leading-strand Tg causes uncoupling of leading-strand synthesis. Since Pol δ outcompetes Pol ϵ for free 3'-ends (Guilliam and Yeeles, 2020, *Nat. Struct. Mol. Biol*) and Tg bypass at higher dNTP concentrations is stimulated in the presence of Pol δ , it is highly likely that a switch to Pol δ during leading-strand synthesis occurs upon uncoupling even when dNTP levels are elevated.

Minor points

1. Page 2, line 15, "We propose that replicase switching promotes continued leading-strand synthesis whenever the replisome encounters leading-strand damage that is bypassed more efficiently by Pol δ than by Pol ϵ ". The last sentence of the abstract is a strong generalization considering the large variety of DNA lesions found in vivo and many factors that can influence the activity of Pol Delta and Pol Epsilon. For example, the authors mention the dNTP concentrations which has not really been investigated in this manuscript.

We have modified this sentence to “We propose that replicase switching **may promote** continued leading-strand synthesis whenever the replisome encounters leading-strand damage that is bypassed more efficiently by Pol δ than by Pol ϵ .” We believe that this is a fair proposal based on the evidence presented in this manuscript and the emerging role for Pol δ in leading-strand synthesis whenever it occurs uncoupled from template unwinding (Guilliam and Yeeles, 2020, *Crit. Rev. Biochem. Mol. Biol.*).

2. Page 17, line 5-7, please rephrase the wording "is likely to" with "may". So far, the studies of three DNA lesions are pointing in that direction but I believe that it is too early to generalize considering the diversity of DNA lesions and also other effectors such as e.g. dNTP concentrations., that may influence the process.

We have made this requested change.

Referee #2:

The paper by Guilliam and Yeeles, entitled, "The eukaryotic replisome tolerates leading-strand base damage by replicase switching", reports on a study of the bypass of leading strand lesions, Tg and 8-oxoG, by the yeast replisome. The authors use a yeast replisome reconstituted with purified proteins, as well as purified pols ϵ and δ to investigate leading-strand lesion bypass. The authors observe that the replisome is tolerant of the Tg and 8oxo-G lesions, and that bypass of these lesions is not dependent on the specialized TLS polymerases, but rather majority of bypass depends on the lagging strand replicase, pol δ . Based on the results the authors conclude that when pol ϵ encounters the lesion it uncouples from the helicase, which permits pol δ to bind, incorporate the correct nucleotide opposite the lesion, and continue synthesis until it catches up with the helicase. At this point a switch to pol ϵ , and its recoupling with the helicase occurs to complete leading strand synthesis.

The results support the authors' conclusions, they are well presented and discussed, and the reasoning is clear. Overall it's a very nice study and a well written paper. It will be of interest to a wide readership of the EMBO Journal.

The authors show that pol ϵ is also able to bypass a template Tg lesion, however bypass by pol δ is more rapid and more efficient than by pol ϵ . It is not clear why this is the case. Could the exonuclease of pol ϵ be responsible for the less efficient bypass

of the Tg lesion, causing idling of the polymerase at the site of the lesion and resulting in its uncoupling from the helicase? Could the authors comment? It would be interesting to perform bypass reactions with an exonuclease deficient variant of pol ϵ . Having said this, I do not suggest that this experiment is necessary for this paper.

We have now compared Tg bypass by Pol ϵ and Pol $\epsilon^{\text{exo-}}$ in a primer extension assay (new Fig EV3E). Interestingly, the predominant stall product in the Pol $\epsilon^{\text{exo-}}$ reactions was shifted from N+10 (stalling immediately before the lesion) to N+12 (stalling upon incorporation of a single nucleotide after the lesion) and some increase in full extension was observed. However, a substantial amount of stalling still occurred compared to reactions containing Pol δ , suggesting the exonuclease activity of Pol ϵ contributes to poor bypass efficiency but is not the only factor responsible for this. We feel that the difference in bypass efficiency between Pol ϵ and Pol δ is a potentially interesting area of future study but further investigations are beyond the scope of the present manuscript, as the referee suggests.

Few additional minor points

Figure 2B, lane 4, there are clear bands at the "stall" and "oligo restart" positions- products of Tg bypass synthesis in the presence of the scrambled oligonucleotide S, why is that? The authors should comment on this.

Please see response to Referee #1 point 4.

Legend to Figure 2A, for clarity, the schematic showing the replication products should be described in more detail; what do the green sections and the dashed line in the synthesized strand represent?

We have now added more detail to the figure legend. The green lines represent the restart oligonucleotides and the dashed line indicates extension of the stall product which occurs due to bypass of the Tg after oligonucleotide binding. This extension up to the 5'-end of the oligonucleotide was observed in our previous study (Guilliam and Yeeles, 2020, *Nat. Struct. Mol. Biol.*).

Page 8, line 10, "On the UD template, FL-lead was generated both in the absence

and presence of Pol δ^{cat} (Fig 2C, lanes 1-6), confirming the mutant does not inhibit coupled leading strand synthesis." However, inspection of Fig 2C suggests that there is less FL-lead product with the UD template in the presence of pol δ^{cat} (compare lanes 1,2,3 with 4,5,6). The same appears to be true in the experiment presented in Fig 6B, again compare lanes 1,2,3, with lanes 4,5,6.

Did the authors quantitate the amount of FL-lead product in these reactions? Is my assessment accurate? If so, please comment why pol δ^{cat} would have an effect in this case.

The key observation from this experiment is that on an undamaged template Pol δ^{cat} does not prevent generation of FL-lead products, but on the Tg template addition of the mutant stalls leading strands at the site of the lesion and completely inhibits FL-lead production. This confirms that all forks have uncoupled at the lesion, allowing Pol δ^{cat} to bind which therefore inhibits lesion bypass/recoupling. The referee is correct that there is a minor decrease in FL-lead products on the UD template when Pol δ^{cat} is present but a significant amount of FL-lead is still generated and this does not therefore affect the overall interpretation of the experiment. This slight decrease in FL-lead on the UD template in the presence of Pol δ^{cat} may be due to two reasons. Firstly, a small minority of replication forks may uncouple before reaching the end of the UD template. Here Pol δ^{cat} could bind the uncoupled nascent leading strand and prevent extension to full-length. Secondly, it is possible that the chase has not worked with 100% efficiency and that a small fraction of forks which have fired after the addition of the chase have been labelled. Addition of Pol δ^{cat} should inhibit initiation of leading strand synthesis (Aria and Yeeles, 2019, *Mol. Cell*), thereby preventing labelling of products in the chase which might occur in a minority of cases in its absence. Importantly, this does not affect the key observation that Tg is causing uncoupling of leading strand synthesis.

Figure 3C, shouldn't the lesion in the template strand be Tg not CPD?

This error has now been corrected.

Referee #3:

This paper investigates the consequence of budding yeast replisome encounters with

damaged DNA bases (thymine glycol [Tg], 8-oxo-G [8oG]) on the the leading strand template. Using the reconstituted budding yeast DNA replication system the authors demonstrate that yeast replisomes are inherently able to bypass Tg or 8oG on the leading strand independent from TLS polymerases. Experimental evidence demonstrates that leading strand synthesis by Pol-epsilon is transiently uncoupled from replisome progression at the site of DNA damage, while re-coupling is mediated after error-free bypass by Pol-delta. The findings highlight the distinct impact of Tg and 8oG on replisome progression from that of CPD damage, as the latter has been demonstrated by the authors previously to be dependent TLS. The data uncover that transient switching between Pol-epsilon and Pol-delta promotes inherent error-free bypass of Tg and 8oG damage sites, thus revealing a further role for the lagging strand polymerase, Pol-delta, in leading strand synthesis.

The data is robust and of high quality, the elegant assays have been previously established by the authors for the study of replisome collisions with CPD damage (Guillian & Yeeles, NSMB, 2020). I recommend publication of this manuscript in its current form.

Thank you for submitting your final revised manuscript for our consideration. I am pleased to inform you that we have now accepted it for publication in The EMBO Journal.

Referee #1:

The authors have addressed all prior questions, have included complementing experiments and made clarifications where needed. I have nothing further to comment as I am satisfied with the response and changes in the manuscript.

Corresponding Author Name: Joe Yeeles

Journal Submitted to: The EMBO Journal

Manuscript Number: EMBOJ-2020-107037